# How Neighborhood Characteristics Influence Neighborhood Crimes: A Bayesian Hierarchical Spatial Analysis

**DOI:** 10.3390/ijerph191811416

**Published:** 2022-09-10

**Authors:** Danlin Yu, Chuanglin Fang

**Affiliations:** 1Department of Earth and Environmental Studies, Montclair State University, Montclair, NJ 07043, USA; 2Center for Urban and Regional Planning Design and Research, Institute of Geographic Science and Natural Resources Research, Chinese Academy of Sciences, Beijing 100045, China

**Keywords:** urban crimes, census block groups, spatial data analysis, Bayesian hierarchical modeling, varying coefficients model, Paterson

## Abstract

Urban crimes are a severe threat to livable and sustainable urban environments. Many studies have investigated the patterns, causes, and strategies for curbing the occurrence of urban crimes. It is found that neighborhood socioeconomic status, physical environment, and ethnic composition all might play a role in the occurrence of urban crimes. Inspired by the recent interest in exploring urban crime patterns with spatial data analysis techniques and the development of Bayesian hierarchical analytical approaches, we attempt to explore the inherently intricate relationships between urban assaultive violent crimes and the neighborhood socioeconomic status, physical environment, and ethnic composition in Paterson, NJ, using census data of the American Community Survey, alcohol and tobacco sales outlet data, and abandoned property listing data from 2013. Analyses are set at the census block group level. Urban crime data are obtained from the Paterson Police Department. Instead of examining relationships at a global level with both non-spatial and spatial analyses, we examine in depth the potential locally varying relationships at the local level through a Bayesian hierarchical spatially varying coefficient model. At both the global and local analysis levels, it is found that median household income is decisively negatively related to urban crime occurrence. Percentage of African Americans and Hispanics, number of tobacco sales outlets, and number of abandoned properties are all positively related with urban crimes. At the local level of analysis, however, the different factors have varying influence on crime occurrence throughout the city of Paterson, with median household income having the broadest influence across the city. The practice of applying a Bayesian hierarchical spatial analysis framework to understand urban crime occurrence and urban neighborhood characteristics enables urban planners, stakeholders, and public safety officials to engage in more active and targeted crime-reduction strategies.

## 1. Introduction

Community safety is one of the primary indicators of a livable and sustainable city [1,2,3,4,5,6,7,8]. Urban space, as the highest spatial structure of human habitation and concentration, harbors great potential for human development and prosperity, but it also serves as a potential hot spot for the dark side of human nature [7,9,10,11,12,13,14,15]. Crime occurrence was never a simple A causes B type of equation but rather involves complex and convoluted socioeconomic [5,16], demographic [15], human psychological [17,18,19], governance [20], and even physical building environmental factors [2,3,6,7,9,21].

Community safety scholars have long explored the relationship between the concentration of alcohol and tobacco sales outlets, ethnic compositions, and urban crime occurrence at county [22,23,24], census tract [12,25,26,27], and census block group [2,28,29,30] levels in the US. The conclusions are clear. Urban crimes tend to occur more often in neighborhoods with more outlets selling harmful products (alcohol and tobacco) and often find their ways in neighborhoods with less than well-to-do socioeconomic status (higher poverty households occupied by minority ethnic groups). This consensus prevails in community and urban criminology literature [3,5,7,12,23,31,32,33,34,35,36] and is the foundation for urban planning, urban crime deterrence, and other urban governance strategies in building better, more livable and sustainable cities in the future.

Traditional analytical methods, while utilizing some of the geographic properties of spatial data (mapping is among the most commonly used geographic properties) to visualize the results, taking full advantage of the spatial effects [37] in data analysis did not mature until very recently [2]. The recent advancement of data analysis technologies thanks to the advanced computational power and innovative algorithms, especially in the field of spatial data analysis, has created a boom in better understanding the underlying mechanisms for many urban social and economic phenomena [2,38,39,40,41,42,43,44]. While many studies capitalize on this newfound analytical prowess for in-depth research [3,4,12,27,35], new approaches and methodological developments continue to push the boundaries for better exploration, better understanding, and better policy support for urban governance, urban crime fighting, and the improvement of urban quality of life and sustainability.

One such recent methodological advancement is the development of Bayesian hierarchical strategy coupled with varying coefficients models [45,46,47]. Bayesian analytical strategies have a long history. The core idea of Bayesian analysis is attractive because the Bayes theorem enables scholars to combine the information carried by the data (observation, the likelihood) and information that was derived from prior experiences, empirical studies, scholarly experiences, or even common sense (the prior) to create a full posterior distribution of the unknown parameters. If no prior experience or empirical studies are available, Bayesian analysis will then rely exclusively on the likelihood, which is equivalent to classical statistical analysis [47,48,49,50]. From the pure theoretical perspective, Bayesian analytical approaches prove to be steady and robust against both data issues and prior issues when carefully specified [47,50]. The primary hurdle to practical Bayesian analysis, however, is that empirical Bayesian analysis relies heavily on the computer-generated simulation of posterior distributions that are the product of likelihood density and prior distribution since analytical solutions are often not attenable in many real-world applications [50]. Bayesian statistical analysis did not gain much attention until the early 2000s, when the Markov Chain Monte Carlo sampling strategy with the Gibbs sampler was developed [51,52,53]. Drastically increased computing power facilitated the spread of Bayesian analysis, especially considering the huge accumulation of prior research experiences on many societal, economic, natural, and physical phenomena. Bayesian analytical approaches are able to tap into this great treasure and potentially produce more versatile and reliable analytical results [50,51].

This premise was especially true in the field of spatial (and later spatiotemporal) data analysis [54,55,56,57]. This is because the Bayesian modeling strategy allows a more flexible research design and scheme that do not have to impose simplified structures that are often elusive in spatial and spatiotemporal data analyses. In addition, spatial/spatiotemporal data analysis has long recognized the spatial effects as inherent characteristics of data collected over geographic spaces [58,59]. The development of spatial data analysis in the decades since the early 1970s [60,61] has attempted to structuralize, model, incorporate, or filter the spatial effects that was the result of the coincidence of locational similarity and attribute similarity [58,62,63,64]. While recognizing that spatial effects have profound impacts on data analysis, especially in regression analysis [37,65,66,67], spatial effects are structuralized with a neighborhood linkage matrix that often has either binary (1 and 0) elements or inversed distance elements. While the structure certainly brings revolutionary understanding of data analysis with geographical properties, it might still be on the simpler side of the spectrum to characterize the full scope and flexibility of the spatial processes that generate the data. Bayesian approach, on the other hand, while taking advantage of the prior knowledge that observations of spatial neighbors are somehow correlated, the correlation structure can be flexibly assumed with hyperparameters and reasonable distributions such as the Gaussian Markov random field [68] that can be effectively used to describe spatial observations that follow certain neighborhood structure.

Bayesian spatial/spatiotemporal analysis thus provides a renewed opportunity for geographers, urban criminologists, spatial epidemiologists, and many other spatial social scientists to hopefully unravel more in-depth understanding of previous problems and generate new knowledge regarding how certain socioeconomic processes operate to produce the data that we observe.

After this brief introduction, we will present our data and detail the Bayesian hierarchical spatial varying coefficients process model. Analytical results are presented in the Section 3, and discussion follows in the Section 4. We conclude our study with a summary and future research foci.

## 2. Materials and Methods

### 2.1. Paterson, New Jersey

The City of Paterson, the county seat of Passaic County, New Jersey, is located in northern New Jersey (Figure 1) approximately 20 miles to the west of New York City. According to the 2020 Census, with a population of 159,732, it is the third most populous city in the state of New Jersey. It has a land area of 8.71 square kilometers according to the US Census Bureau, which makes it the fourth most densely populated city in the US.

Paterson initially served as the cradle of the American industrial revolution in the early 19th century. The nearby Great Fall provide ample hydraulic power to encourage the development of industrial mills and factories. The booming of Paterson’s industries made the city an ideal location for new immigrants. In the latter half of the 19th century, Paterson’s silk production dominated the local economy and earned the city the nickname “Silk City.” Since then, it has become a major destination for Hispanic immigrants as well as immigrants from Turkey, the Middle East, and South Asia. In the 2020 census, it was estimated that there are anywhere between 75 to 100 different languages spoken in the city. In the early half of the 20th century, Paterson developed into a booming industrial center and urban area.

The declination of urban area and ensuing suburbanization after the end of the second world war gradually brought decline to the city of Paterson. Beginning in the 1960s, the unemployment rate in Paterson started to climb, and many residents had to move out of the city to find jobs. While new immigrants continued to come to the city, especially Hispanic immigrants, the post-industrialization economic mode that emphasized the knowledge economy and information technology, as well as continuous suburbanization and exurbanization [69,70,71], brought further decline to the city of Paterson that continues today. While aging residents of the city often remember the glorious days the city once enjoyed, with mounting unemployment rate and uncertain economic cycles, the city has suffered much more than simple economic downturn. As a side product, the crime rate within the city has picked up its pace. Establishments such as alcohol and tobacco sales outlets that were often associated with shadowy operations started to gain ground within the city. In addition, because residents were moving out of the city, many properties were also abandoned, creating a gloomy landscape that is more than dotted eyesores (Figure 1). Of the six wards in Paterson, Wards 1, 4, and 5 are particularly infested with violent crimes from the available 2012 crime data we gathered.

Mapping the distribution of violent crime enables policy makers and city officials to have a clear picture of the crime landscape. For better governance and for creating and maintaining a vibrant and livable, sustainable city, understanding, investigating, and exploring the fundamental urban elements that might be associated with crime occurrence are critical.

### 2.2. Violent Crimes, Harmful Products, Urban Prosperity, and Ethnic Landscape

The declination of a city is not something that happens overnight; there are fundamental causes that are rooted in the society, the physical environments, the economy, and even the ethnic landscape [2,27,72]. The occurrence of urban crimes almost certainly ensues from the declination of cities; the abandonment of properties; and the loss of family-building, community-bonding, and future-promising employment opportunities [4,7,10,14,15,25,28,34,73,74,75,76,77]. The increased crime occurrence will in turn facilitate further declination of the city, entering the city into a vicious cycle. Without clear understanding of where to break this cycle, the city and its residents could suffer from countless negative consequences such as degraded infrastructure, loss of business, loss of property value, higher unemployment rate, loss of population, and ill-maintained cityscape and building environments, among many others. In addition, while not explicitly stated or even consciously aware, outlets that sell harmful products, such as tobacco and alcohol, tend to increase their presence in declining neighborhoods [12,78,79,80]. All these factors combined cause a city to lose its vibrancy and decrease its livability in the long run.

This study collected various assaultive violent crime occurrences in the City of Paterson from the Paterson Police Department, which includes both aggravated assaults and robbery, from the year 2013. Actual locations where the violence occurred were geocoded and then integrated (via spatial join in a GIS) to the census block groups to produce the count information within each block group; then, a heat map was produced to show the distribution of crime in the city (Figure 1). Sociodemographic factors, which include the total population, percentage of Hispanic population, percentage of African American population, and median household income within the census block group, were obtained from the New Jersey Department of Labor, and the US Census Bureau. Tobacco (355) and alcohol (197) sales outlets information were obtained from the Paterson Alcohol Beverage Control Board (2013). The actual addresses of those outlets were geocoded and integrated (via spatial join in a GIS) to the census block group as well. Abandoned property information for 2013 was acquired from Paterson’s Housing Authority (1557), and a similar GIS operation was conducted to integrate the count to the census block group level. The map of tobacco and alcohol sales outlets and abandoned properties and the heat map of violent crimes are presented in Figure 1. A visual inspection shows clear spatial associations between urban crime/livability and the numbers of tobacco and alcohol sales outlets and abandoned properties. Still, a formal analysis is required to provide confirmation of the relationships among these factors.

### 2.3. Bayesian Hierarchical Modeling Approach

In many previous works, investigating the relationships between crime and socioeconomic and sociodemographic characteristics, urban infrastructure, and harmful product sales outlets is usually carried out with either ordinary least squares regression [12,27,81] or simultaneous spatial autoregressive models [2,82,83], or, often in the case of investigating spatially varying relationships, geographically weighted regression or eigenfunction-based spatial filtering [79,82]. Among these approaches, the spatial approaches (either spatial autoregression or spatially varying regression) are often preferred since the data are collected over geographic space that is governed by the First Law of Geography [84] and the spatial effects of both autocorrelation and heterogeneity tend to dominate the data-generating process.

In addition to these commonly applied spatial data analysis and modeling techniques, the recent development of Bayesian statistics has lent power to spatial data analysis. Approaches that are based on the Bayes theorem provide viable alternatives for looking at the relationships among variables. The Bayesian analytical framework proceeds as follows. First, through the observed data *Y* (violent urban crime counts per census block group), we can establish a likelihood function that specifies the distribution of the data under the model determined parameters (θ):$$L\left(\theta \right)=p(Y=y|\theta )$$

Since the violent urban crimes per census block group are count data, a Poisson family distribution is well suited for establishing the likelihood. Further experiments with the data suggest that a negative binomial distribution represents the data’s distribution well and is adopted as the likelihood distribution.

Second, in our current study, we are interested in the relationships between the inputs, namely, socioeconomic and demographic characteristics, harmful product sales outlets, and physical built environments, and the output, violent urban crimes. These relationships are conveniently modeled through a regression analysis and presented as the coefficients of the inputs. Since the data are collected over space, we could separate the regression residual into an unstructured residual and a spatially structured residual [55]. In this case, the coefficients are assumed to stay constant over space. We have accumulated an extensive amount of prior knowledge on spatial structure-determined random effects such as identified by Besag and colleagues [56,85]. The details are extensively discussed in [54,55,57,85] and will not be repeated here.

If, however, we assume that the unknown parameters do not necessarily stay constant over space, as seen in many local analyses (such as GWR and other spatially varying coefficient models), then instead of separating the residuals, a hierarchical structure with each observation within a geographical unit as the first level (individual observation level) and the geographical location as the second level (group level) can be specified. In this hierarchical structure, although each group (the geographic location) contains only one observation, we can assume that each observation is sampled from a different population that is specific to the geographic location (group). While it is possible to perform a stratified analysis within the Bayesian analytical framework by fully specifying a probability distribution as the prior (for instance, a non-informative prior such as $\theta_{j}\sim Normal\;\left(0,\;10000\right)$ can be specified), since we only have one sample in each group (geographic location), the estimation for θ is going to be highly variable and unstable. In addition, this assumption violates the First Law of Geography because the First Law dictates that at the group level (geographic location level), $\theta_{j}$s are somehow dependent on one another based on their geographic locations (closer things are more related than distant ones). This prior knowledge (First Law of Geography) allows us to take full advantage of the hierarchical structure of the data because $\theta_{j}$s are no longer considered independent; instead, they are assumed to come from one single distribution $p(\theta_{j}|\psi )$ characterized by the same hyperparameter ψ, and we have accumulated sufficient prior knowledge of this hyperparameter ψ because of the First Law of Geography.

We are replacing the spatial structure with a hierarchical structure. The advantage of doing so is that each group (geographic location *j*) will now have its own unique parameter ($\theta_{j}$) because the observation in each location is assumed to be sampled from different populations. This unique parameter, however, is similar to other observations’ parameters because the hierarchical structure is spatial in nature and can be characterized based on the accumulated knowledge of spatial dependence with hyperparameters [54,55] as p(θ|ψ), where ψ is the hyperparameter. This modeling structure is the so-called varying coefficient model under the Bayesian framework [47]. This model assumes a hierarchical structure for each parameter as a unique spatial modifier to create a spatially varying coefficient process model (local model) [55]. The model is specified as:$$y_{i}=f\left(\eta_{i}\right)=\overset{P}{\underset{\;p=1}{\sum }}\left(\beta_{p}+\beta_{pi}\right)x_{pi}$$
where $y_{i}$ is the outcome variable (violent crime occurrence) at census block group *i*. $f\left(\eta_{i}\right)$ is the link function, which follows a negative binomial specification for count data. *p* is the number of predictors, including the intercept. $x_{pi}$ is the *p*th covariate (the socioeconomic, demographic, and physical built environment characteristics) at location *i*. $\beta_{p}$ is the average coefficient of the *p*th covariate. $\beta_{pi}$ is the space effect modifier for the *p*th covariate at location *i* (*i* = 1, …, *n*). The coefficients and spatial modifiers are the parameters (θ) we are interested in estimating.

To estimate the parameters, we need to specify their priors so that using the data generated likelihood distribution, we will be able to produce the posterior distributions. The average coefficients are called fixed effects and are typically normally distributed, centered on 0 with large variances [55]. For the spatial effect modifier, if the variation is considered discrete, as in our current study in which the data are collected over areal units (census block groups), it is considered a Gaussian Markov random field (GMRF) [68] and can be modeled with an intrinsic conditionally autoregressive (ICAR) model [47,85].

A GMRF is a very intuitive graphic representation of the First Law of Geography that is also based on a predefined neighborhood structure. In a GMRF, let the spatial modifier ${\left(\beta_{p1},\ldots ,\;\beta_{pn}\right)}^{T}$ have a normal distribution with mean ***µ*** and covariance matrix ***Σ***. Define the labelled graph $G=\left(V,\;E\right)$, where $V=\left\{1,\;\ldots ,\;n\right\}$ is the set of vertices (geographic areas) and *E* is a set of edges such that there is no edge between vertices *i* and *j* if these two vertices are not considered neighbors (predefined neighborhood structure). Then we say that the spatial modifier ${\left(\beta_{p1},\ldots ,\;\beta_{pn}\right)}^{T}$ is a GMRF with regard to *G*. It turns out that the labelled graph, *G*, determines the matrix structure of the precision matrix, ***Q***, which is the inverse of the covariance matrix. In the precision matrix, the elements *Q_ij_* is non-zero only when spatial units *i* and *j* are considered neighbors. In our study, we use the sphere of influence (SOI) rule to determine the neighborhood structure and generate the labelled graph *G*. For a sphere of influence (SOI) for the *i*th unit, let *r_i_* be the distance from *i* to its nearest neighbor and *C_i_* the circle centered on *i* with the radius of *r_i_*; *i* and *j* are SOI neighbors when *C_i_* and *C_j_* intersect. With this definition, the spatial effect modifier can then be considered a GMRF with regard to *G* that is generated based on the SOI rule of the census block groups of Paterson.

The ICAR model for all the *p* covariates at location *i* indicates the prior of the spatial effect modifier can be expressed as:$$\beta_{i}|\beta_{-i}\sim N\left(\frac{1}{n_{i}}\underset{j:i\sim j}{\sum }\beta_{j},\;{\left(n_{i}\tau \right)}^{-1}\right)$$
where $\beta_{-i}$ is the coefficient on locations other than *i*; $n_{i}$ is the number of neighbors of location *i*; and i~j denotes locations *i* and *j* are neighbors as identified in the graph *G*. τ is a precision parameter (the hyperparameter) [47].

In addition, to avoid potential overfitting problems, we also adopt the penalized complexity priors for all the hyperparameters (the precisions of the varying coefficients) [47,86]. When specifying the penalized complexity priors for the hyperparameters, we found that the ICAR model can be seen as a random walk of order one model with rank equals to *n*–1 (*n* being the number of spatial units). Based on the discussion by Simpson, Rue, Riebler, Martins, and Sørbye [86], we assumed a relatively small standard deviation (0.3) as an upper bound for the spatial random effects (the varying effects) to reflect our expectation that the neighborhood conditions have relatively similar effects on crime occurrence. To test the sensibility of the models to this hyperparameter, we used an even smaller standard deviation (0.1) and a relatively large standard deviation (0.5) to calibrate the model. The results do not change much, suggesting that the model is robust against the hyperparameter. With the prior specified, our model is complete. The model is calibrated with the recently proposed integrated nested Laplace approximations (INLA) algorithm in R platform [55,87], which is comparable with but much faster than the commonly employed MCMC algorithm for Bayesian statistics simulation.

## 3. Results

Under the Bayesian framework, we first establish a generalized linear model between the violent crime counts and the demographic factors (total population, *Pop*, percentage of Hispanics, *pcthisp*, percentage of African Americans, *pctaa*), socioeconomic (median household income, *MHI*), and physical built environment (tobacco, *tbc* and alcohol, *alc* sales outlets and abandoned properties, *abdp*) using the negative binomial distribution [2]. Three models are estimated under the Bayesian framework using R-INLA. The first model is a regular negative binomial regression without specifying any spatial effects as the random effects. The second model specifies the spatial effects as structured random effects and adopts a penalized complexity prior based on the Besag specification for the spatial structure [85]. The third model adopts penalized complexity priors for all the coefficients under the varying coefficient process modeling framework.

All three models are calibrated in R [88] with the spdep package [89] and R-INLA package [48]. The spedp package was used to create the sphere of influence (SOI) neighborhood structure. The R-INLA package converted the neighborhood structure to a graph to build the Gaussian Markov random field for the spatial structure and incorporate the spatial structure in the spatial and spatially varying coefficient models. For both models, the penalized complexity priors for the spatial structures are adopted to avoid potential model overfitting. The saturated deviance information criteria (saturated DIC) are reported in R-INLA and are used for model comparison, with smaller values suggesting a preferred model [48,55].

The results for Models 1 and 2 are reported in Table 1 and Table 2. Table 3 reports the summary of the estimated varying coefficients and corresponding varying *t*-values.

In addition, based on the varying *t*-values, we are also able to identify for the spatially varying coefficients model, total population, median household income, number of tobacco sales outlets, percentage of African Americans, and number of abandoned properties show some significant spatially varying patterns. These patterns are mapped in Figure 2 to demonstrate the spatial variations in the relationships between violent crime occurrence and demographic, socioeconomic, and physical built environment characteristics at the census block group level.

## 4. Discussion

Reading the tables and figures, we have some interesting results to share with the urban livability and sustainability community. First, from looking at the three tables, an immediate impression emerges. The spatial model with penalized complexity prior and the non-spatial model report similar results. The saturated DICs of both models are also very similar (911.0929 for the nonspatial model and 910.4023 for the spatial model). This observation is contrary to previous findings [2,80,83] with similar model structures and calibrations. In general, classical spatial data analysis techniques that are heavily dependent on the spatial autocorrelation structure of the regression residuals tend to suggest that the existence of spatial autocorrelation in the regression residuals will violate the regression assumptions and that thus, maximum likelihood estimators will produce viable alternatives, and often it turns out the alternatives perform better than ordinary least squares estimation. While this is interesting at first glance, the comparison is not fair to start with. This is because estimation with ordinary least squares with spatially autocorrelated residuals is not valid, and given this, comparison between a valid estimator (the maximum likelihood estimator) and an invalid estimator (the ordinary least squares estimator) might not tell the full story. Under the Bayesian analytical framework, however, since the estimation does not depend on the minimization of the squared residuals (instead, spatial structure is introduced as a random effect in the model structure, and a penalized complexity prior for the random effect is utilized), we argue that the comparison might be more reasonable.

From the comparison of the spatial and nonspatial models under the Bayesian analytical framework, we contend that at the census block group level, the spatial structure of the census block groups does not have a significant impact on the modeling results when added as a separate random effect. This result suggests that when conducting spatial analysis, we need to take caution before we claim that spatial effects always cause discernible differences from their non-spatial counterparts, as is often reported in spatial autoregressive analysis. Both models suggest that violent crimes tend to occur more often in more populated block groups with lower median household incomes, higher concentrations of ethnic minorities, harmful product sales outlets, and dilapidated neighborhood environments. This result is not different from many of the previous studies’ findings [2,7,14,15,24,35,90,91,92]. The result is also one of the cornerstones for municipal governance and policing practices.

Second, while adding the spatial structure as an additional random effect to the regression model does not yield significant differences from the regular regression analysis within the Bayesian analytical framework, adding spatial structures separately to each of the explanatory variables yields a better model fit (the saturated DIC is 896.7343) and suggests more interesting relationships between the factors and urban crime occurrence at the individual census block group level (instead of the collective census block group level as in the previous model). This result is interesting not only because it agrees with a widely observed result that local models often fit the data better even after the added complexity compared with global models [82,93,94,95] but also because it suggests that spatial structure might very well be more of an individual structure that is inherently embedded with individual explanatory variables over space than a collective one. By ignoring the spatial structure of individual explanatory variables but attempting to capture the spatial structure via an added random effect, even with the same Gaussian Markov random field, and penalized complexity priors for the hyperparameters, spatial effects at the collective level might still fail to present or exert significant impact on modeling performance or results. While practitioners of varying coefficient modeling, especially the popular geographically weighted regression analysis, often focus on the varying coefficients and the model’s added flexibility and detail-explaining power [80,94,95], scholars often do not explain why local spatial models account for the spatial effects better than global spatial ones. The current exploration with the urban crime analysis offers a possible alternative explanation that spatial structure and spatial effects are likely more of an individual structure that is unique to each individual explanatory variable instead of a global effect that can be modeled collectively, or at least sufficiently modeled collectively. This finding is critical in that it provides solid theoretical guidance for analyzing data collected over geographic space. While considering spatial structure and spatial effects first as a collective structure that can be modeled as a random effect, it is imperative to model individual structure and effects to produce a holistic understanding of the relationships presented within the model. This is especially critical for analyzing urban crimes and what factors affect urban crimes at detailed municipal levels because we will be using the results as policy guidance for urban governance, crime fighting, policing practices, and building a livable and sustainable urban future. More reasonable and hence more reliable modeling results will provide solid support for effective policies and actions to fight urban crime and create a smart, more livable, and sustainable urban future.

Third, with the above modeling understanding, our focus now turns to the results presented by the varying coefficients model that are mapped in Figure 2. Not surprisingly, crime occurrence is generally high in places where there are more people. This pattern, however, is more salient in Wards 1, 4, and 5, where crime occurrence is more concentrated. In the other three wards, the numbers of people in the census block groups do not have significant impacts on local crime. This can be explained by the “pockets of crime” theory [96]. Wards 1, 4, and 5 of Paterson are infested with violent crimes, which suggests these wards possess certain characteristics that might offer advantages to violent crimes. More dense populations only exacerbate these advantages and lead to more violent crimes. In the other wards where such advantageous characteristics for violent crimes do not prevail, population concentration does not seem to be related to the occurrence of violent crimes. Our Bayesian hierarchical spatial modeling results can verify that the pockets of crime theory clearly manifests here in Paterson, NJ.

Two advantageous characteristics for violent crime occurrence in these wards are the concentration of tobacco and alcohol sales outlets and the number of abandoned properties (Figure 1). As a matter of fact, while in the local model, the numbers of alcohol sales outlets and abandoned properties at the census block group levels do not present widespread significant associations with the occurrence of violent crime (Table 3), the number of tobacco sales outlets shows a rather consistent pattern over Wards 1, 4, and 5. While the lack of significant local relationships between alcohol sales outlets and abandoned properties seems to be contradictory to previous findings [2,27,82] and also to the global model’s results (Table 1 and Table 2), we are not particularly surprised. At the global modeling level, tobacco and alcohol sales outlets and abandoned properties at the census block group level in Paterson do not show significant multicollinearity (none of the VIFs exceeds 4), a careful examination of Figure 1 shows a clear spatial overlapping pattern of these three urban landscape elements. While under the Bayesian analytical framework in the current study, we do not examine the local multicollinearity among the explanatory variables, we suspect that at the local level, tobacco and alcohol sales outlets and abandoned properties have high levels of spatial overlapping, which leads to the lack of significant relationships between violent crime occurrences and the other two landscape elements.

The lack of significant relationships with the concentrations of Hispanic population as suggested by the global model, is likely for the same as above. The City of Paterson has a dominant Hispanic population. According to the US Census Bureau population estimates as of 1 July 2021, 61.4% of Paterson’s residents are Hispanic or Latino. The prevalence of Hispanic population might very well create a local multicollinearity situation between percentage of Hispanic residents and total population. A quick re-run of the varying coefficient model without the total population as one of the explanatory variables produces the t-value for percentage of Hispanic residents ranging from 1.863 to 1.995 with a mean of 1.936 instead of the reported range in Table 3 (1.527–1.652 with a mean of 1.601).

Significant but varying relationships between the violent crime occurrence and median household income and percentage of African American residents at the census block group level are well established in the literature [2,73,97,98,99]. Neighborhoods with disadvantageous socioeconomic status are often called hotbeds for crime [36,100]. In general, in American society, African Americans often have disadvantageous societal and economic status compared with other ethnic groups and are often taken advantage of by harmful product sales outlets [83,101,102,103]. As a result, concentrations of African Americans at the census block group level are universally positively related with higher occurrences of violent crimes. This relationship prevails in all six wards in Paterson, with slightly stronger relationships in Wards 1, 2, 4, and 5. The spatial varying pattern of the median household income at the census block group level, however, demonstrates an almost supplementary pattern to that of the total population. In Wards 1, 4, and 5, the median household income and violent crime occurrence show weak or even no significant relationships, while in the other wards, higher median household income often suggested lower violent crime occurrence (Figure 2). This pattern can also be explained by the ”pockets of crime” theory [96]. In the violent crime-laden Wards 1, 4, and 5 (the pockets of crime), the disadvantageous characteristics such as high concentrations of harmful product sales outlets and abandoned properties (Figure 1) dominate the occurrence of violent crime. Within these pockets, variations in income are no longer sufficient to bring variations in the occurrence of violent crimes. In other wards, however, income level suggests the socioeconomic landscape, which again demonstrates the relevant relationship between income and violent crime occurrence: higher incomes are associated with less crime.

## 5. Conclusions

The current study reexamined the relationship between violent crime occurrence and demographic, socioeconomic, and physical built environment factors in Paterson, New Jersey, at the census block group level. We performed the analysis with Bayesian hierarchical spatial models. While the global level of analysis produces similar results to those often discussed in the literature, the local level analysis suggests two important take-home messages. First, spatial structure is important when analyzing urban crime occurrence and its potential contributing factors. The varying coefficient model under the Bayesian hierarchical modeling scheme, however, suggests that spatial structure is better modeled with individual explanatory variables instead of as a collective whole, especially considering that the priors for the hyperparameters that characterize the spatial structures are penalized based on their complexity. Not only will the varying coefficient model provide a more detailed picture of the investigated relationships, the model is also more realistic when integrating spatial effects in data analysis.

Second, while the occurrence of crime is never an easy phenomenon that can be tackled by finding the causes and stopping them, analyzing the data, especially with a holistic investigation as introduced in the current study, does provide solid urban governance and crime fighting supports. For instance, while many might argue that improving incomes might prevent more violent crimes, our study suggests that changing income levels might not work as well if the fundamental physical built environment of the neighborhood is not changed first. Similarly, while the model indicates that concentrations of African Americans residents might be inconveniently related with higher occurrence of violent crimes, it is not the concentration of any particular ethnic group but again, the fundamental and structural socioeconomic disparity that leads to this relationship landscape. Still, the patterns presented in the current study also serve as urban policing guidance for strategic planning and responding to possible future crimes.

Third, the successful application of the Bayesian hierarchical spatial model framework in this study provides a fresh methodological approach to urban studies that involve data collected over geographic units. When any geographic unit is treated as its own hierarchy but connected under the dominance of the First Law of Geography, an informed prior that takes advantage of this knowledge (spatial autocorrelation) for the hyperparameter of the coefficient could produce more reasonable and potentially more reliable estimates. Future studies that consider relationships among variables collected over geographic space can take advantage of the proposed approach.

Urban crime is no doubt one of the biggest threats to a smart, livable, vibrant, and sustainable city. Fighting crime, however, requires not only effective and efficient policing but more importantly, understanding the root of the problem and being able to deal with the problem from the fundamental level. Our reexamination of the crime occurrence and demographic, socioeconomic, and physical built environment factors in Paterson, New Jersey, clearly suggests that urban crime is the surface manifestation of the deeper problem of social injustice. While policing and urban renovation projects could mitigate the problem to certain degrees, a smart, livable, vibrant, and sustainable city needs to address the fundamental roots of social and economic injustice. Only then will cities and city dwellers be able to enjoy the city life and landscapes all together.

## Figures and Tables

**Figure 1 ijerph-19-11416-f001:**
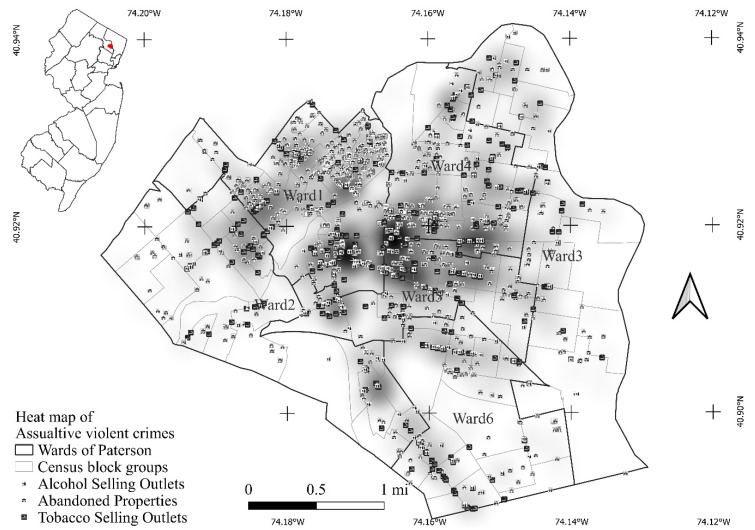
Location of Paterson, NJ, and assaultive violent crime heat map.

**Figure 2 ijerph-19-11416-f002:**
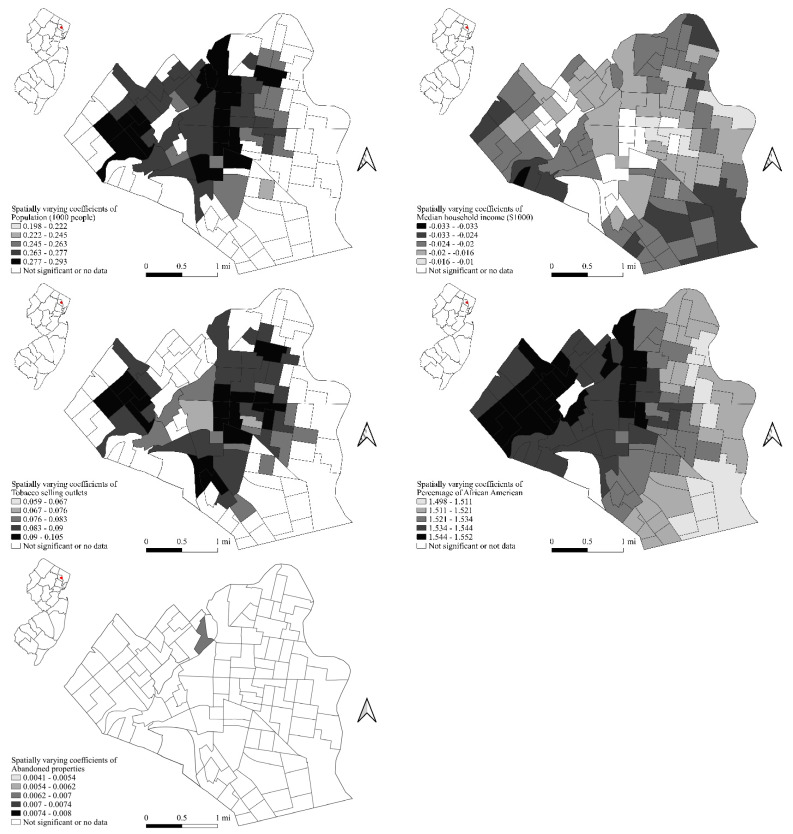
Spatially varying coefficients.

**Table 1 ijerph-19-11416-t001:** Nonspatial negative binomial regression results.

Variables	Mean	sd	0.025 Quant	0.5 Quant	0.975 Quant
(Intercept)	0.602	0.435	−0.255	0.604	1.453
*Pop*	0.220	0.098	0.030	0.218	0.416
*MHI*	−0.021	0.004	−0.028	−0.021	−0.014
*pcthisp*	1.590	0.480	0.649	1.589	2.534
*pctaa*	1.995	0.415	1.183	1.994	2.813
*alc*	0.052	0.037	−0.020	0.052	0.125
*tbc*	0.103	0.021	0.062	0.103	0.144
*abdp*	0.005	0.002	0.000	0.005	0.010

Saturated DIC: 911.0929.

**Table 2 ijerph-19-11416-t002:** Spatial negative binomial regression results (using the Besag structure with penalized complexity prior).

Variables	Mean	sd	0.025 Quant	0.5 Quant	0.975 Quant
(Intercept)	0.584	0.447	−0.301	0.586	1.457
*Pop*	0.219	0.098	0.031	0.218	0.414
*MHI*	−0.021	0.004	−0.028	−0.021	−0.014
*pcthisp*	1.602	0.493	0.635	1.601	2.574
*pctaa*	2.018	0.438	1.163	2.016	2.886
*alc*	0.053	0.037	−0.020	0.053	0.126
*tbc*	0.101	0.021	0.061	0.101	0.143
*abdp*	0.005	0.002	0.000	0.005	0.010

Saturated DIC: 910.4023.

**Table 3 ijerph-19-11416-t003:** Summary of the spatially varying estimated coefficients and t-values (with the Besag spatial structure for each variable and penalized complexity prior).

Variable	N	Mean	Std. Dev.	Min	Pctl. 25	Pctl. 75	Max
*Pop.b **	105	0.256	0.023	0.198	0.244	0.274	0.293
*Pop.t ***	105	1.912	0.252	1.243	1.793	2.09	2.324
*MHI.b*	105	−0.019	0.004	−0.033	−0.021	−0.017	−0.01
*MHI.t*	105	−2.427	0.446	−3.331	−2.776	−2.123	−1.28
*pcthisp.b*	105	1.121	0.02	1.081	1.104	1.136	1.161
*pcthisp.t*	105	1.601	0.029	1.527	1.585	1.625	1.652
*pctaa.b*	105	1.531	0.014	1.498	1.519	1.542	1.552
*pctaa.t*	105	2.252	0.029	2.187	2.231	2.271	2.317
*tbc.b*	105	0.083	0.009	0.059	0.078	0.09	0.105
*tbc.t*	105	1.973	0.405	1.048	1.68	2.214	2.993
*alc.b*	105	0.066	0.014	0.029	0.057	0.076	0.107
*alc.t*	105	1.079	0.248	0.359	0.916	1.235	1.649
*abdp.b*	105	0.007	0.001	0.004	0.006	0.007	0.008
*abdp.t*	105	1.129	0.24	0.643	0.969	1.242	2.045

*: *b* represents the estimated coefficients. **: *t* represents the calculated *t*-values for the estimated coefficients. Saturated DIC: 896.7343.

## Data Availability

Data and analytical scripts will be available upon reasonable request.

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
