# Peer review of "How Neighborhood Characteristics Influence Neighborhood Crimes: A Bayesian Hierarchical Spatial Analysis"

_ijerph, 2022, doi:10.3390/ijerph191811416_

Round 1
Reviewer 1 Report
Dear authors,
I believe that the front-end of the paper is comprehensive and well-written. I want to praise the author's extreme effort and work in undertaking the exploration of urban crime patterns using the Bayesian hierarchical analytical techniques. However, there are some concerns related to the methodology and the study findings.
In terms of the significant contribution to the field, the paper tries to play a significant contribution, but I was not able to glean the results in a way that is applicable to other studies or future work in the field.
According to the title of the manuscript, the purpose of the study is the spatial configuration of neighbourhood crime. Configurational analysis consists of a number of methods to measure both the links connecting the spaces through which people move in the urban environment. The metrics of spatial configuration, including those devised by the technique called Space Syntax Analysis, provide useful variables that can be introduced as regressors in spatial configuration analysis. I am wondering if the authors can link the space syntax to the Bayesian hierarchical models in this study. A study conducted by di Bella and colleagues in 2015 has suggested the linkage between different statistical methods of urban crime (i.e., space syntax and Bayesian hierarchical framework). Or at least, I would like to see some comparisons between these two types of statistical methods of urban crime.
In conclusions, I would like to see much more discussion of the links between the findings and implications for theory, policy and practice. There is some general discussion of the implications, but this is broad and not enough.
Author Response
Dear authors,
I believe that the front-end of the paper is comprehensive and well-written. I want to praise the author's extreme effort and work in undertaking the exploration of urban crime patterns using the Bayesian hierarchical analytical techniques. However, there are some concerns related to the methodology and the study findings.
Response: Thank you for your kind words and insightful review. We will attempt to address your concerns to the fullest and revise our manuscript to the better.
In terms of the significant contribution to the field, the paper tries to play a significant contribution, but I was not able to glean the results in a way that is applicable to other studies or future work in the field.
Response: Thank you for this significant comment. When we are applying the current approaches to study urban crime patterns and the broad urban health issues, we had this in mind that the application of Bayesian Hierarchical analytical approaches will be of great importance for other types of analyses that utilize similar spatial data sets. We did not state this intention explicitly. We added one take-home message in the summary section to explicitly reflect this point.
According to the title of the manuscript, the purpose of the study is the spatial configuration of neighbourhood crime. Configurational analysis consists of a number of methods to measure both the links connecting the spaces through which people move in the urban environment. The metrics of spatial configuration, including those devised by the technique called Space Syntax Analysis, provide useful variables that can be introduced as regressors in spatial configuration analysis. I am wondering if the authors can link the space syntax to the Bayesian hierarchical models in this study. A study conducted by di Bella and colleagues in 2015 has suggested the linkage between different statistical methods of urban crime (i.e., space syntax and Bayesian hierarchical framework). Or at least, I would like to see some comparisons between these two types of statistical methods of urban crime.
Response: Thank you for the comment. Our title attempts to construct a relationship between neighborhood characteristics and neighborhood crimes, from a Bayesian hierarchical spatial analysis perspective. We should be more specific about that. For that matter, we altered the title to “How neighborhood characteristics influence neighborhood crimes: a Bayesian hierarchical spatial analysis.” As for the space syntax analysis, we are not familiar with the approach and feel inadequate to compare the model with the Bayesian hierarchical spatial model in the current study, though by visiting di Bella and colleagues’ work, we do feel the space syntax analysis has great potential to add to the Bayesian hierarchical spatial analysis framework. We thank you for this interesting comment and will be exploring this opportunity in the future.
In conclusions, I would like to see much more discussion of the links between the findings and implications for theory, policy and practice. There is some general discussion of the implications, but this is broad and not enough.
Response: We provide more detailed discussion especially in regard to how the findings are related with policies and practices, as well as urban criminology theory development in the fourth section (Discussion). We hope the revision will gain your approval.
Reviewer 2 Report
This paper is a very interesting research but will improved using a more specfic case of study, which include for example a Krigging model as in:
Xinyan Li, Lei Miao:
The Study on Spatial Distribution of Floor Area Ratio Based-on Kriging - - The Case of Wuhan City. GRMSE 2014: 538-547
and
https://www.springerprofessional.de/en/intelligent-system-for-predicting-motorcycle-accident-by-reachin/17923920
Is very important an analyze very descriptive with multivariable análisis to this research.
In addition, is very important make a comparative with related research in a real world problems in a Smart City.
Many situations associated in this moment with violence are provoked by the confinement and isolation of citizens.
In many largest city the situation is very caotics in this moment because the econom,y affect to the people, is important describe with HDI Index the map in this research.
Author Response
This paper is a very interesting research but will improved using a more specfic case of study, which include for example a Krigging model as in:
Xinyan Li, Lei Miao:
The Study on Spatial Distribution of Floor Area Ratio Based-on Kriging - - The Case of Wuhan City. GRMSE 2014: 538-547
and
https://www.springerprofessional.de/en/intelligent-system-for-predicting-motorcycle-accident-by-reachin/17923920
Is very important an analyze very descriptive with multivariable análisis to this research.
Response: Thank you for your comments and recommendation to the above studies that apply Kriging interpolation approaches to study Floor Area Ratio and Motorcycle accidents. These are interesting studies from a spatial interpolation perspective, and we appreciate your comments. Our study, on the other hand, focuses on applying a Bayesian hierarchical spatial modeling framework to study the influences of urban neighborhood characteristics on urban crimes. The concept of “Smart City” is of critical importance since one of the aims of the current study is to achieve a “smart, livable, and sustainable” urban future. We added comments about “smart city” in the discussion and hope the revision will gain your approval.
In addition, is very important make a comparative with related research in a real world problems in a Smart City.
Response: Thank you for your comments. When describing the methodology of Bayesian hierarchical spatial modeling framework and also in the discussion section, we have made quite a few comparisons with previous studies that utilize different techniques to study urban crimes and urban characteristics (for instance, section 2.2, section 4). We do not specifically use the term “smart city” because we feel the expression of “livable, sustainable” city has largely overlapping the connotation of “smart city.” We hope this clarification could gain your approval, but thank you for your comments, nonetheless.
Many situations associated in this moment with violence are provoked by the confinement and isolation of citizens.
In many largest city the situation is very caotics in this moment because the econom,y affect to the people, is important describe with HDI Index the map in this research.
Response: Thank you for your comments. This is very true for now after the world has suffered and endured a very long global pandemic. Isolation and chaos dominated many parts of the world for quite some time. However, our current study is an example of applying Bayesian hierarchical spatial modeling frameworks with data in 2013, while confinement and isolation certainly contribute to occurrence of violent crimes even then, but we do not have means to collect relevant data at the census block group level the degrees of confinement and isolation. We also appreciate that you mentioned the Human Development Index for urban crime fighting and smart city construction. Again, HDI is not readily available at the census block groups level, but the components that are used to measure HDI, life expectancy, quality of life, and knowledge, are somehow overlapping with median household income, ethnic composition and landscapes of harmful products and abandoned properties. While we appreciate your insightful comments, we hope our clarification will gain your approval of our revision.